# EMPIRICAL STUDY OF EASY AND HARD EXAMPLES IN CNN TRAINING

## ABSTRACT

Deep Neural Networks (DNNs) generalize well despite their massive size and capability of memorizing all examples. There is a hypothesis that DNNs start learning from simple patterns based on the observations that are consistently well-classified at early epochs (i.e., *easy examples*) and examples misclassified (i.e., *hard examples*). However, despite the importance of understanding the learning dynamics of DNNs, properties of easy and hard examples are not fully investigated. In this paper, we study the similarities of easy and hard examples respectively among different CNNs, assessing those examples contributions to generalization. Our results show that most easy examples are identical among different CNNs, as they share similar dataset-dependent patterns (e.g., colors, structures, and superficial cues in high-frequency). Moreover, while hard examples tend to contribute more to generalization than easy examples, removing a large number of easy examples leads to poor generalization, and we find that most misclassified examples in validation dataset are hard examples. By analyzing intriguing properties of easy and hard examples, we discover that the reason why easy and hard examples have such properties can be explained by biases in a dataset and Stochastic Gradient Descent (SGD).

## 1 INTRODUCTION

From a traditional perspective of generalization, overly expressive models can memorize all examples and result in poor generalization. However, deep neural networks (DNNs) achieve an excellent generalization performance even if models are over-parameterized (Zhang et al., 2017). The reason for this phenomenon remains unclear. (Arpit et al., 2017) show that brute-force memorization does not happen in DNNs training, and propose a hypothesis that DNNs start learning from simple patterns. Their hypothesis is based on observations of examples that are consistently well-classified at early epochs (i.e., easy examples) and examples misclassified (i.e., hard examples). However, despite their importance in understanding how DNNs start learning, easy and hard examples are not thoroughly examined.

In this paper, we study easy and hard examples, and their intriguing properties are shown. First, we investigate whether easy and hard examples differ on various CNNs architectures. For our experiments, we introduce *easiness* as a metric to measure how early examples are classified. According to *easiness*, we calculate the matching rates of easy and hard examples between different CNN architectures. Surprisingly, CNNs start learning from identical easy examples and they are visually similar to each other. It indicates that different CNNs start learning from similar patterns.

Second, we investigate the patterns that make examples *easy*. We study CNNs with examples from which some patterns are intentionally removed. We find that the patterns CNNs start learning from are dataset dependent (e.g., colors, structures, and high-frequency information).

Lastly, we conduct ablation and corrupted-label experiment to determine how easy and hard examples contribute differently to generalization. We find that hard examples contribute more to generalization than easy examples, however, removing a large number of easy examples leads to poor generalization. This phenomenon can be explained by biases in a dataset. In addition, we discover that misclassified examples in validation dataset are consistent and mostly hard examples.

Those properties can not be explained by a hypothesis that CNNs start learning from simple patterns (Arpit et al., 2017). From observations, we hypothesize that CNNs start learning from frequent patterns that are not contradicted across classes. There are two reasons why there are easy and hard examples: biases in a dataset and Stochastic Gradient Descent (SGD).

A dataset naturally contains various biases leading some patterns to appear as a majority or a minority. The majority and minority patterns are called *frequent patterns* and *rare patterns*, respectively. Since SGD randomly picks samples to train a model, frequent patterns tend to be used more than rare patterns in a batch. It means that the gradients of frequent patterns dominate the direction of the update, and some examples that are discriminative by frequent patterns are consistently well classified at early epochs.

## 2 APPROACH

In this paper, we perform several experiments to investigate unknown properties of easy and hard examples. We introduce *easiness* as a metric of how early examples are classified. Based on *easiness*, we reveal several properties of easy and hard examples.

### 2.1 EASINESS

To measure how early an example is classified, we introduce *easiness* $e_t$ as a criteria, where $t$ denotes the index of an example. We repeat $m$-epochs training $N$ times and take a record of the number of correct classifications. Let $c_t$ be the number of trials in which the example $x_t$ is correctly classified. *easiness* $e_t$ is computed as:

$$e_t = \frac{c_t}{\max_{1 \leq j \leq K} c_j},$$ 
(1)

where $K$ is the total number of examples, and $e_t$ is in the range of $0 \leq e_t \leq 1$, with 0 indicating that $x_t$ is the hardest examples and with 1 showing that $x_t$ is the easiest examples.

We then redefine easy and hard examples as follows:

**Easy examples** and **Hard examples**: $\{x_t | e_t > \tau, t \in [1, K]\}$ and $\{x_t | e_t < \tau, t \in [1, K]\}$.

In our experiments, $\tau$ is calculated so that 10% of examples with the highest and lowest *easiness* belong to easy and hard examples, respectively.

### 2.2 MATCHING RATE

Based on *easiness*, we use matching rate to measure how similar easy and hard examples are between various CNN architectures. Let us consider two different set of examples $X_A$ and $X_B$. The matching rate $M_{AB}$ between $X_A$ and $X_B$ is calculated as:

$$M_{AB} = \frac{|X_A \cap X_B|}{\max(|X_A|, |X_B|)},$$ 
(2)

where $|\cdot|$ denotes the size of a set.

## 3 EXPERIMENTS

### 3.1 PREPARATIONS

We utilize CIFAR-10 (Krizhevsky, 2009) and ImageNet 2012 dataset (Russakovsky et al., 2015) for our experiments.

**CIFAR-10.** On CIFAR-10, translation by 4 pixels, horizontal flipping, and global contrast normalization are applied onto images with $32 \times 32$ pixels. We use a list of models to CIFAR-10: three layer

Table 1: The matching rates of easy and hard examples among various CNNs in CIFAR-10 and mini ImageNet. Since the values in the tables are symmetric and redundant, we put "−" on unnecessary table cells for clarity. We use 10% of examples with the highest and lowest *easiness* to calculate the matching rate between different CNNs architectures. Random means that we randomly select 10% of examples from the datasets.

|  | MLPs | WRN | DenseNet-BC | ResNeXt | Random |
|---|---|---|---|---|---|
| MLPs | 0.70 | 0.20 | 0.19 | 0.21 | 0.1 |
| WRN | - | 0.86 | 0.71 | 0.49 | 0.1 |
| DenseNet-BC | - | - | 0.91 | 0.47 | 0.1 |
| ResNeXt | - | - | - | 0.82 | 0.1 |

(a) Matching rate of easy examples in CIFAR-10

|  | MLPs | WRN | DenseNet-BC | ResNeXt | Random |
|---|---|---|---|---|---|
| MLPs | 0.96 | 0.15 | 0.14 | 0.14 | 0.1 |
| WRN | - | 0.90 | 0.64 | 0.50 | 0.1 |
| DenseNet-BC | - | - | 0.83 | 0.47 | 0.1 |
| ResNeXt | - | - | - | 0.86 | 0.1 |

(b) Matching rate of hard examples in CIFAR-10

|  | MLPs | AlexNet | VGG | VGG-BN | ResNet | DenseNet | Random |
|---|---|---|---|---|---|---|---|
| MLPs | 0.63 | 0.61 | 0.67 | 0.31 | 0.52 | 0.48 | 0.1 |
| AlexNet | - | 0.87 | 0.72 | 0.41 | 0.50 | 0.48 | 0.1 |
| VGG | - | - | 0.90 | 0.37 | 0.59 | 0.59 | 0.1 |
| VGG-BN | - | - | - | 0.87 | 0.25 | 0.31 | 0.1 |
| ResNet | - | - | - | - | 0.96 | 0.65 | 0.1 |
| DenseNet | - | - | - | - | - | 0.75 | 0.1 |

(c) Matching rate of easy examples in mini ImageNet

|  | MLPs | AlexNet | VGG | VGG-BN | ResNet | DenseNet | Random |
|---|---|---|---|---|---|---|---|
| MLPs | 0.0 | 0.07 | 0.03 | 0.0 | 0.01 | 0.01 | 0.1 |
| AlexNet | - | 0.94 | 0.48 | 0.09 | 0.26 | 0.25 | 0.1 |
| VGG | - | - | 0.78 | 0.20 | 0.45 | 0.26 | 0.1 |
| VGG-BN | - | - | - | 0.46 | 0.46 | 0.42 | 0.1 |
| ResNet | - | - | - | - | 0.52 | 0.44 | 0.1 |
| DenseNet | - | - | - | - | - | 0.68 | 0.1 |

(c) Matching rate of hard examples in mini ImageNet

multilayer perceptrons (MLPs) WRN 16-4 (Zagoruyko & Komodakis, 2016), DenseNet-BC 12-100 (Huang et al., 2017) and ResNeXt 4-64d (Xie et al., 2017)

**mini ImageNet.** We randomly select 10 classes[1] from ImageNet 2012 dataset for an interpretable analysis and call them *mini ImageNet*. On mini ImageNet, resizing images with the scale and aspect ratio augmentation and horizontal flipping are applied onto images. Then, global contrast normalization is applied to randomly cropped images with $224 \times 224$ pixels. We use a list of models to mini ImageNet: three layer MLPs, AlexNet (Krizhevsky et al., 2012), VGG11(Simonyan & Zisserman, 2015), VGG11 with Batch Normalization (Ioffe & Szegedy, 2015), ResNet 18 (He et al., 2016) and DenseNet 121 (Huang et al., 2017)

As the optimizer, we use AdamW (Loshchilov & Hutter, 2017) and parameters are as follows: $\alpha = 0.001$, $\beta_1 = 0.9$, $\beta_2 = 0.99$, $w_{norm} = 0.05$ and $T_i = 300$. We train a model for 300 epochs and use 128 batch size for CIFAR-10 and 64 batch size for mini ImageNet.

---

[1]Image IDs are n01751748, n02123394, n02169497, n02883205, n03125729, n03954731, n04332243, n04355358, n04553703 and n07717556.

## 4 Easy examples are identical in different CNNs

To determine whether easy and hard examples are shared across different CNN architectures, we calculate matching rates according to *easiness*.

**Easy examples.** The matching rates of easy examples between CNNs in Table 1 (a) and (c) are much higher than the case of random selection. This result shows that **CNNs start learning from the same examples even if CNN architectures are different**. In addition, the matching rate between MLPs and CNNs are low in CIFAR-10, but high in mini ImageNet.

**Hard examples.** The matching rates of hard examples are shown in Table 1 (b) and (d). Intriguingly, in mini ImageNet, the matching rates among CNNs with batch normalization (Ioffe & Szegedy, 2015) are high. It shows that batch normalization in CNNs architecture affects hard examples. Since easy examples are mostly identical, the difference of hard examples relates to the difference of generalization ability. It requires further analysis.

**Why there are easy and hard examples?** We hypothesize that CNNs start learning from frequent patterns that are not contradicted across classes. It can be explained by biases in a dataset and SGD.

As a example, let us consider the classification problem of a ladybird and mantis. If a ladybird and mantis are always onto a leaf in training examples, the pattern of a leaf is harmful to discriminate examples. We call such harmfulness as contradiction. SGD force the model not to use contradicted patterns. If the model uses the pattern of a leaf to discriminate a mantis, ladybird examples stochastically update the model to discriminate a ladybird by the pattern of a leaf, and then the model start misclassifying mantis. On the one hand, classified and misclassified examples by contradicted patterns interrupt to use contradicted patterns each other. On the other hand, non-contradicted patterns across classes are learned without interferences and used to discriminate examples. When examples are classified by non-contradicted patterns, their loss value and the magnitudes of their gradients become smaller. Thus, the model stops learning contradicted patterns.

There are biases in a dataset, thus some patterns appear frequently and some patterns appear rarely. Since SGD updates the model by averaged gradients, the gradients of frequent patterns become global directions of the gradient to update the model. Easy examples are discriminative by frequent and non-contradicted patterns, thus those patterns are learned first. As easy examples are learned and get smaller loss values, rare patterns in hard examples start being learned. That's why examples are identical among different CNN architectures since such patterns are dependent on a dataset and not on CNNs architectures.

## 5 Easy examples are visually similar to each other

Visualizations[2] of easy and hard examples are shown in Figure 1. To calculate *easiness*, WRN 16-4 and ResNet18 are respectively used for CIFAR-10 and mini ImageNet. As can be seen in Figure 1, easy examples are visually similar to each other, and hard examples tend to be visually different from each other. This result implies that **different CNNs start learning from similar patterns**.

## 6 Why CNNs consider those examples as *easy*

To further analyze what properties make examples *easy* for CNNs, we perform three experiments: color, shuffle and radial masking experiment (Jo & Bengio, 2018). In these experiments, we calculate easiness based on examples from which some patterns are intentionally removed.

**Color experiment.** To investigate whether CNNs look at colors at early epochs, we use easy examples that are interpolated to grey. Let us consider that $R$, $G$, $B$, and $Grey \in \mathbb{R}^{H \times W}$ are red, green, blue and grey channels respectively with height $H$ and width $W$. The interpolations are calculated as follows:

$$R_\alpha = \alpha R + (1.0 - \alpha)Grey,$$
$$G_\alpha = \alpha G + (1.0 - \alpha)Grey, \quad (3)$$
$$B_\alpha = \alpha B + (1.0 - \alpha)Grey,$$

---

[2]The rest of easy and hard examples are provided in the supplementary material.

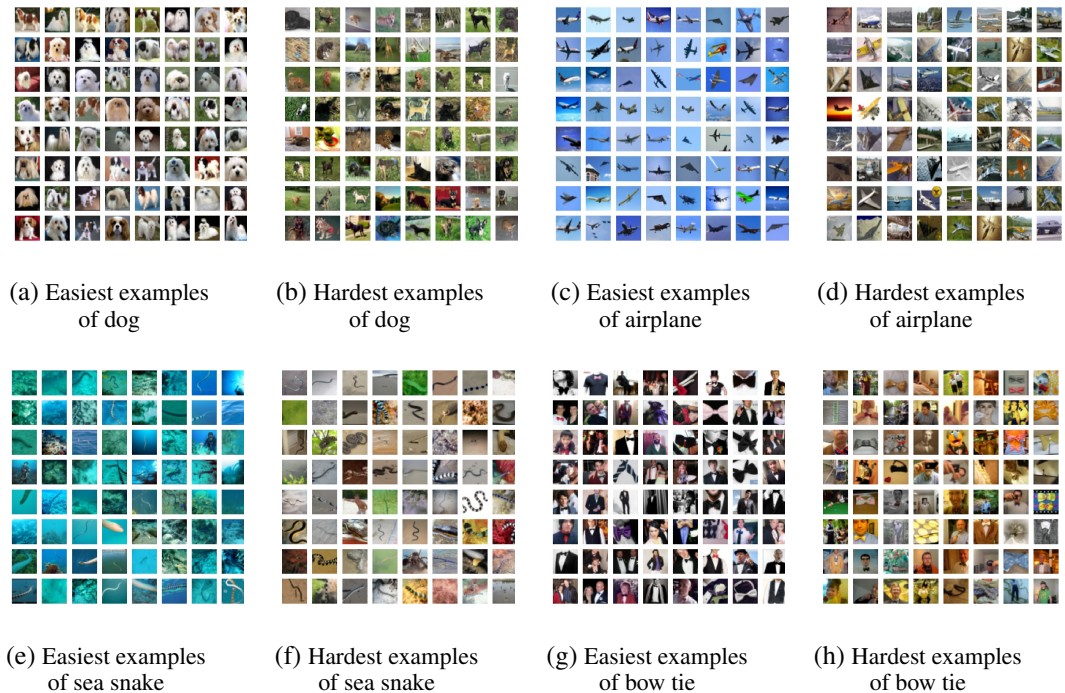

(a) Easiest examples
of dog

(b) Hardest examples
of dog

(c) Easiest examples
of airplane

(d) Hardest examples
of airplane

(e) Easiest examples
of sea snake

(f) Hardest examples
of sea snake

(g) Easiest examples
of bow tie

(h) Hardest examples
of bow tie

Figure 1: Easy and hard examples of CIFAR-10 dataset and mini ImageNet. **a)-d)** are examples from CIFAR-10. **e)-h)** are examples from mini ImageNet.

where $\alpha$ is in the range of $0 \leq \alpha \leq 1$, with 0 indicating that the example is the grey picture and with 1 showing that the example is colored. $Grey$ channel is obtained by averaging $RGB$ channels: $\frac{R+G+B}{3}$. The interpolated examples are shown in Figure 3 (a).

**Shuffle experiment.** The work of (Noroozi & Favaro, 2016) inspires the idea of shuffle experiment. This experiment lets us understand whether CNNs look at structures at early epochs. Then, we cut images into pieces with fixed tile size and reconstruct images with randomly shuffled tiled images. The number of correct classification for the example $c_t$ is calculated by using shuffled tiled images. Examples of the shuffled images are depicted in Figure 3 (b). Tile Size means the relative size when compared to the full image size (e.g., if Tile Size is 0.33, the size of a tiled image is the 33% size of the full image.).

**Radial masking experiment.** (Jo & Bengio, 2018) show that DNNs misclassify examples without high-frequency information that a human can recognize, and claims that DNNs solve problems by looking at superficial cues on examples. The goal of this experiment is to check whether CNNs look at such superficial cues to classify easy examples at early epochs. The work of radial masking experiment is conducted by (Jo & Bengio, 2018). Radial mask $M_r \in \mathbb{R}^{H \times W}$, where $H$ and $W$ are height and width, is computed as follows:

$$M_r[i,j] = \begin{cases} 1 & \text{if } \left\| (i,j) - (\frac{W}{2}, \frac{H}{2}) \right\|_{L_2} \leq r, \\ 0 & \text{otherwise,} \end{cases}$$

where $i$ and $j$ are indices of $M_r$. $M_r$ is a radial mask in the Fourier domain that removes higher-frequency in examples. The bigger $r$ is, the higher frequency of an example is kept. Radial masked examples are shown in Figure 3 (c).

**Result and analysis.** To verify what patterns CNNs start looking at, we calculate the matching rate between regular examples and examples from which some patterns are intentionally removed. First, we train a model for $m$-epochs with regular data augmentation. Then, easiness is calculated by using examples from which some patterns are intentionally removed.

The result are shown in Figure 2. The result of color experiment is the opposite tendency to the result of shuffle and random masking experiment. In CIFAR-10, CNNs start looking at the structure and high-frequency information. In mini ImageNet, CNNs start looking at the colors. It indicates that initially learned patterns are dataset dependent.

In mini ImageNet, we observe that the accuracy does not drop when $r$ is 1.1 small in radial masking experiment. As radial masked examples are depicted in Figure 3 (c), such blurred examples are not recognizable. It implies that CNNs may learn some pseudo patterns if they make sense in training examples. In addition, the trained model with regular data augmentation (after 300 epochs training) consistently well classify some unrecognizable blurred examples (10 out of 10 trained ResNet 18). Such learned pseudo patterns remain into the model to the end.

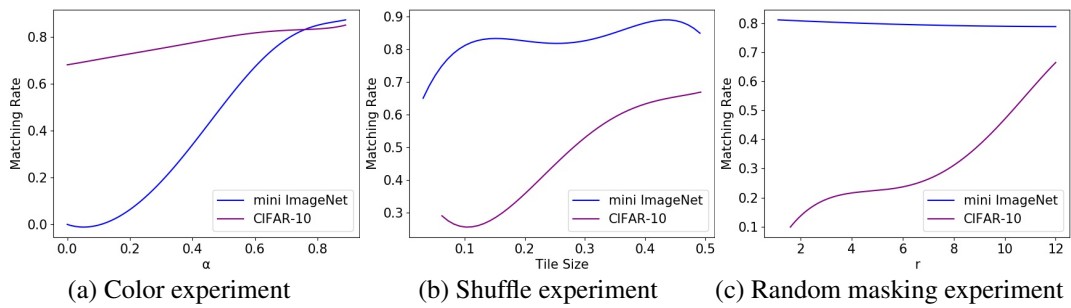

(a) Color experiment     (b) Shuffle experiment     (c) Random masking experiment

Figure 2: The result of easy examples in (a) color, (b) shuffle and (c) radial masking experiments in CIFAR-10 and mini ImageNet. The vertical axis is the matching rate between the case with regular examples and examples from which some patterns are intentionally removed. The model is trained for $m$-epochs with regular data augmentation.

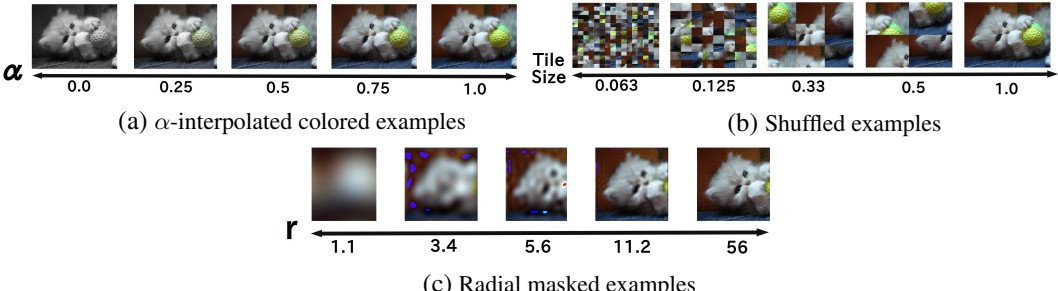

(a) $\alpha$-interpolated colored examples     (b) Shuffled examples

(c) Radial masked examples

Figure 3: Colored, shuffled and radial masked examples.

## 7   EASY AND HARD EXAMPLES CONTRIBUTE DIFFERENTLY TO GENERALIZATION

**Ablation experiment.** We perform ablation experiment on easy and hard examples to determine if they equally contribute to the generalization ability. Before training, we sort training examples according to *easiness* and decide which to discard by beta distribution $Beta(\alpha, \beta)$. When we mainly remove easy examples, $Beta(\alpha = 3, \beta = 1)$ is used. In the same manner, $Beta(\alpha = 1, \beta = 3)$ is used for ablations of hard examples and $Beta(\alpha = 1, \beta = 1)$ is used for the random case.

The result is shown in Figure 4 (a) and (b). As can be seen, removing hard examples consistently degrades the classification performance. Intriguingly, if we remove many easy examples, the accuracy starts degrading sharply around the drop rate is 0.6. Randomly removing examples consistently produces the best performance.

Frequent patterns in easy examples are redundant, so the accuracy does not degrade when some easy examples are dropped. However, rare patterns in hard examples are unique and removing them result in the drop of the accuracy. Interesting properties is that the accuracy starts degrading sharply when

we remove many easy examples. This phenomenon can be explained by biases in a dataset. Since we randomly split a dataset into training and validation, biases and tendency of easy examples are shared. Thus, if the model cannot classify easy examples in training dataset, the accuracy is dropped sharply. Easy examples are the majority and they strongly affect the accuracy.

**Corrupted labels experiment.** We conduct the experiment of corrupted labels (Zhang et al., 2017) on easy and hard examples. Before training, we sort training examples according to *easiness* and determine which to put random labels by beta distribution $Beta(\alpha, \beta)$. $Beta(\alpha = 3, \beta = 1)$, $Beta(\alpha = 1, \beta = 1)$ and $Beta(\alpha = 1, \beta = 3)$ are used for the case of easy examples, the case of hard examples and the random case, respectively.

The result is shown in Figure 4 (c) and (d). The accuracy of the case of easy examples starts degrading around random label rate 0.3 and a model can not learn in heavy noise case (e.g., the situation in 90% of examples are wrongly labeled.). However, the accuracy of the case of hard examples degrades consistently and a model can learn in heavy noise case.

This phenomenon can be also explained by biases in a dataset. Putting wrong labels onto many easy examples results in an interruption of learning frequent patterns. If many examples are wrongly labeled and a model does not learn frequent patterns, the classification performance is dropped since frequent patterns appear frequently in validation examples. That's why the accuracy starts degrading sharply when many easy examples are wrongly labeled, and the case of hard examples maintain the accuracy in heavy noise case because some easy examples are kept in training examples.

## 7.1 Do misclassified examples in validation dataset are hard examples?

The answer is yes. To calculate *easiness* of examples in validation dataset, we merge training examples with examples in validation and calculate train CNNs for $m$ epochs. Then, we calculate easiness by using validation examples.

The result is shown in Figure 5. Surprisingly, mostly misclassified examples are hard examples. We believe that some *rare patterns* are necessary to classify such misclassified examples and those *rare patterns* are none or a few in training examples.

## 8 Related work

A dataset naturally contains various biases. For instance, (Ponce et al., 2006) shows some averaged images of Caltech-101 (Li et al., 2004) are not homogeneous and recognizable. They claim that Caltech-101 may have *inter-class* variability but lacks *intra-class* variability. (Torralba & Efros, 2011) mentions several biases in a dataset: *Selection bias* means that examples in a dataset tend to have particular kinds of images (example: there are many examples of a sports car in the car category). *Capture bias* represents the manner in which pictures are usually taken (example: a picture of a dog is usually taken from the front with the dog looking at the photographer and occupying most of the picture). In this paper, we find that easy and hard examples are highly associated with such biases.

There are three types of training schemes that start learning from easy examples (i.e., curriculum learning)(Kumar et al., 2010), uncertain examples (Settles, 2010; Chang et al., 2017) or hard examples (i.e., hard negative mining) (Lin et al., 2017; Simo-Serra et al., 2015; Shrivastava et al., 2016; Loshchilov & Hutter, 2016; Wang & Gupta, 2015). Those methods rely on the loss value for each training example, thus properties of easy and hard examples are implicitly used. For examples, when hard examples are emphasized, rare patterns are emphasized and frequent patterns are de-emphasized, thus imbalanced problems in *intra-class* and *inter-class* are mitigated.

(Stabinger & Rodriguez-Sanchez, 2017) suggest an active learning method based on *RDE* (quite similar to our *easiness*). They actively take samples from hard examples. The result of Figure 4 also shows that actively taking a sample from hard examples is an efficient way to increase the classification performance. We believe active sampling on rare patterns is a promising way.

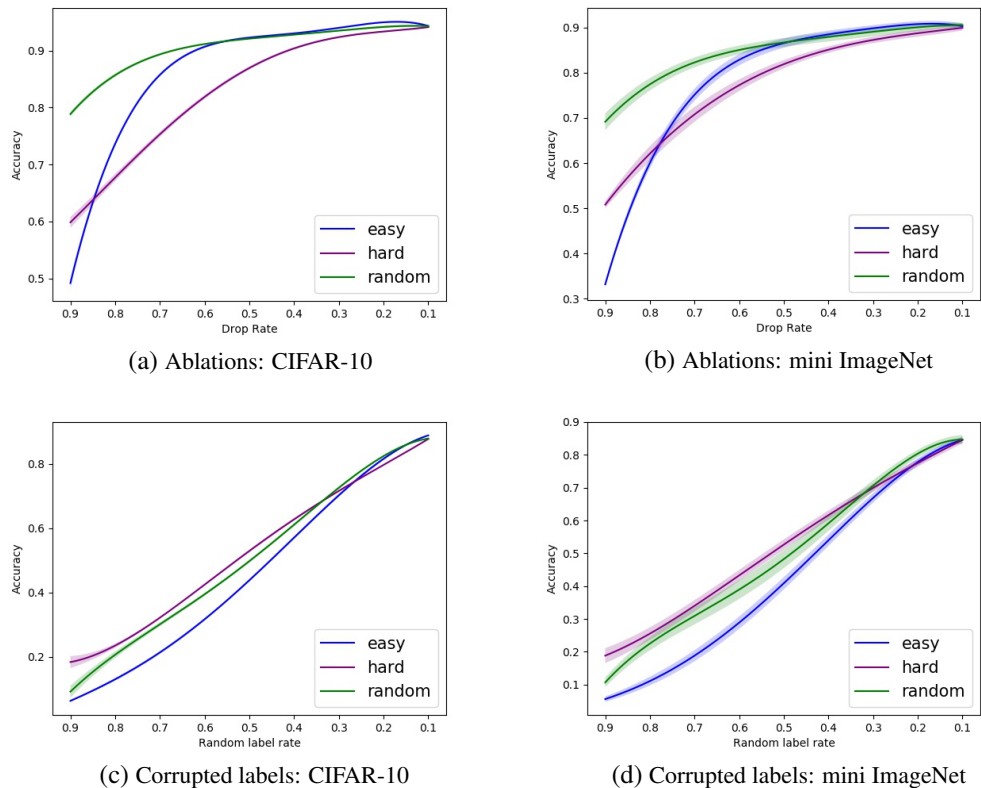

(a) Ablations: CIFAR-10    (b) Ablations: mini ImageNet

(c) Corrupted labels: CIFAR-10    (d) Corrupted labels: mini ImageNet

Figure 4: The result of ablation and corrupted label experiment. The vertical axis is the accuracy. The horizontal axis is the drop rate or random label rate. If the drop rate is 0.3, it means that 30% of examples are discarded. If random label rate is 0.3, it means that 30% of examples are randomly labeled. WRN 16-4 and ResNet 18 are used respectively for CIFAR-10 and mini ImageNet. "easy" on figures means that examples with high *easiness* are mainly removed or randomly labeled. In the same manner, "hard" denotes examples with low *easiness* are mainly processed and "random" represents processed examples are randomly selected. Error bars represent standard deviation over 5 trials.

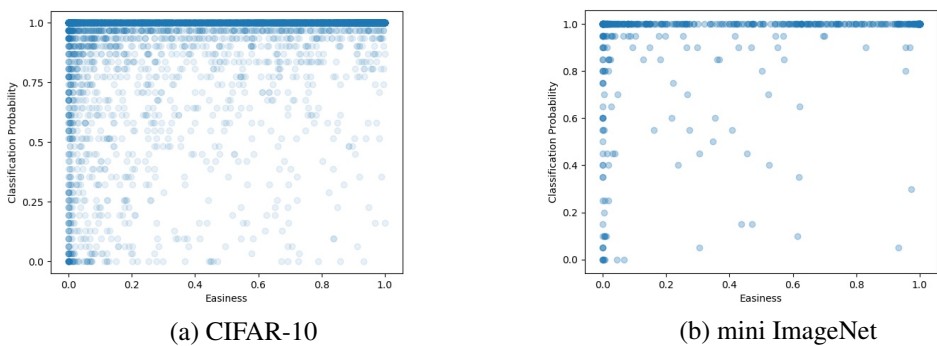

(a) CIFAR-10    (b) mini ImageNet

Figure 5: The relation between easiness and classification probability of validation examples. The horizontal axis is *easiness* of validation examples. The vertical axis is the average of classification probability of validation examples, with 0 indicating that the example is misclassified every time and with 1 showing that the example is classified correctly every time. **a**) Result in CIFAR-10. To calculate classification probability, we take an average of the outputs of 15 trained WRN 16-4 and 15 trained ResNeXt29 4-64d. **b**) Result in mini ImageNet. To calculate classification probability, we take an average from the output of 10 trained ResNet 18 and 10 trained DenseNet 121.

# 9 CONCLUSION AND FUTURE WORK

In this paper, unknown properties of easy and hard examples are revealed. Our experiments demonstrate that different CNNs start learning from identical examples, and easy and hard examples contribute differently to generalization. However, discovered properties can not be explained by a hypothesis that CNNs start learning from simple patterns (Arpit et al., 2017). Consequently, to explain why easy and hard examples have such properties, we hypothesize that CNNs start learning from frequent patterns that are not contradicted across classes.

As future work, by using *easiness*, dataset compression and mislabel mining can be considered for a large-scale dataset. Besides, we would like to analyze the phenomenon of easy and hard examples on natural language and sound domain. Our work utilize datasets with 10 classes, thus it is necessary to conduct further research on large-scale and more complexed datasets.

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

## 10 VISUALIZATION OF EASY AND HARD EXAMPLES

Easy and hard examples of CIFAR-10 and mini ImageNet are shown in Figure 6, 7.

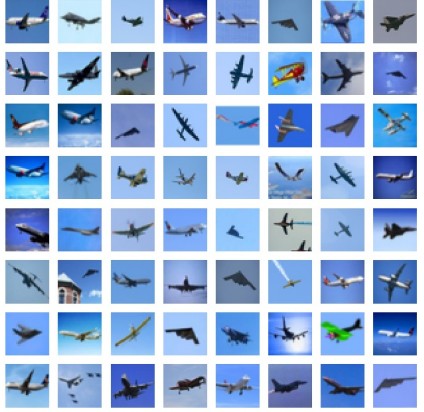

(a) Easiest examples of airplane

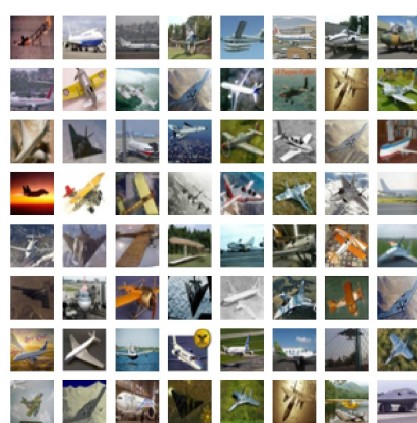

(b) Hardest examples of airplane

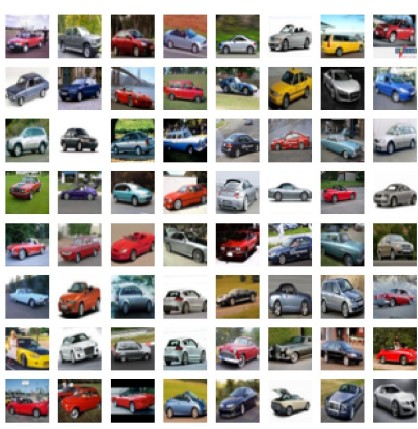

(c) Easiest examples of car

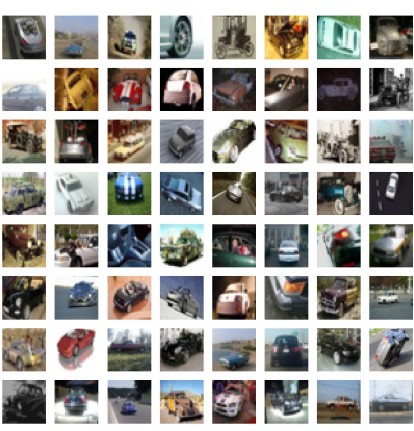

(d) Hardest examples of car

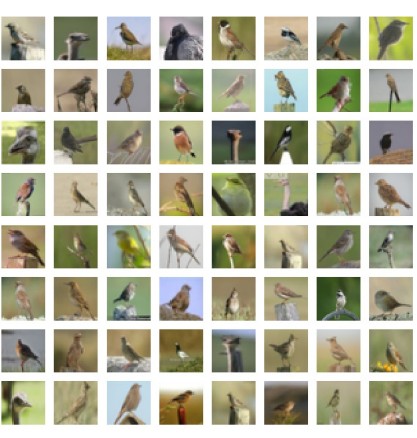

(e) Easiest examples of car

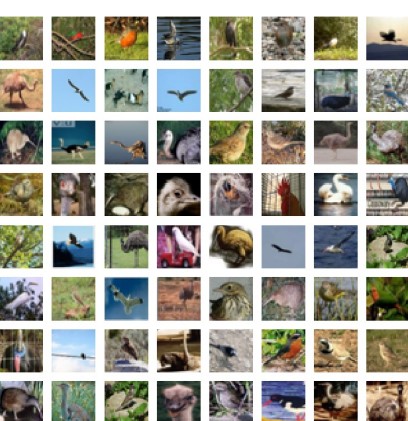

(f) Hardest examples of car

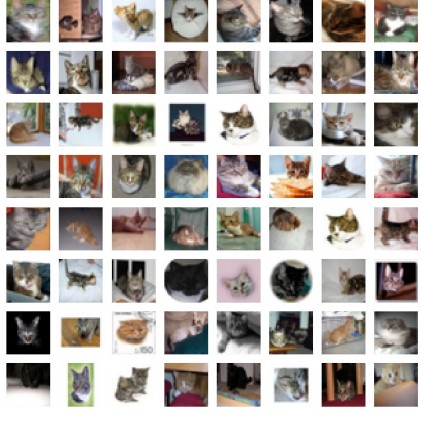

(g) Easiest examples of cat

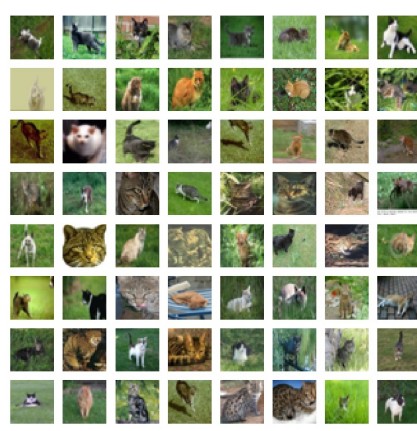

(h) Hardest examples of cat

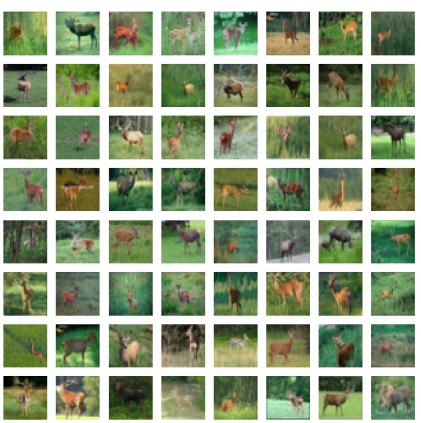

(i) Easiest examples of deer

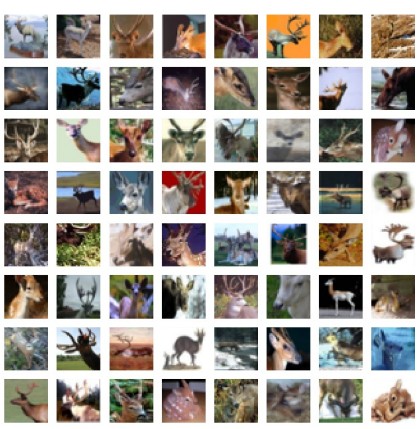

(j) Hardest examples of deer

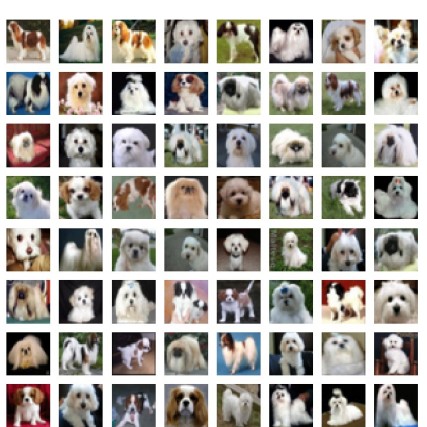

(k) Easiest examples of dog

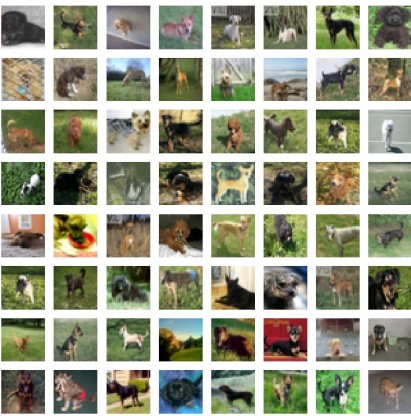

(l) Hardest examples of dog

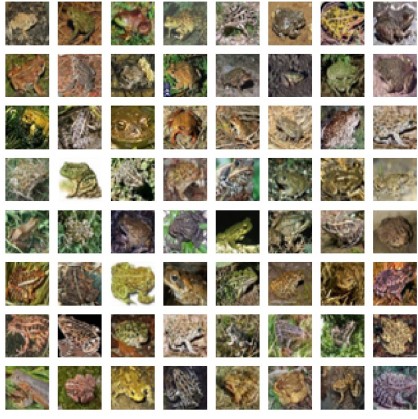

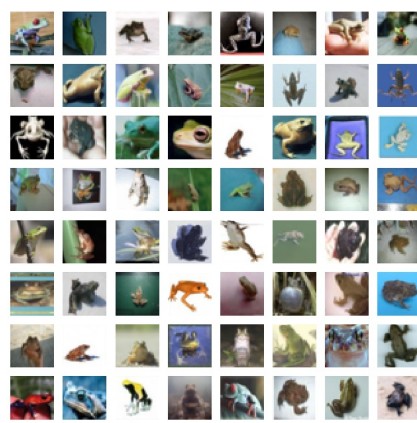

(m) Easiest examples of frog

(n) Hardest examples of frog

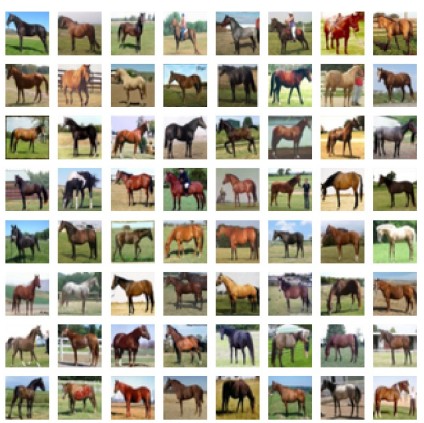

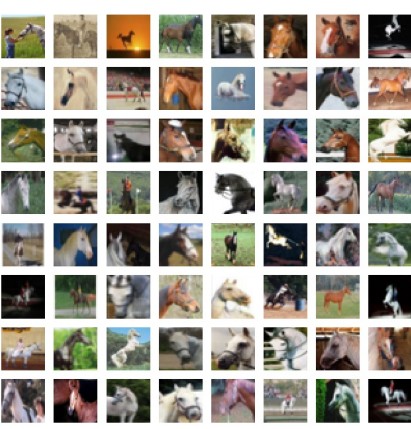

(o) Easiest examples of horse

(p) Hardest examples of horse

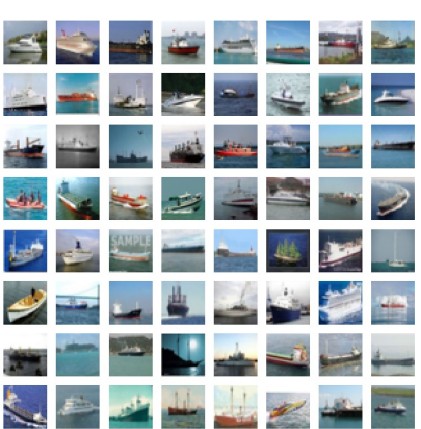

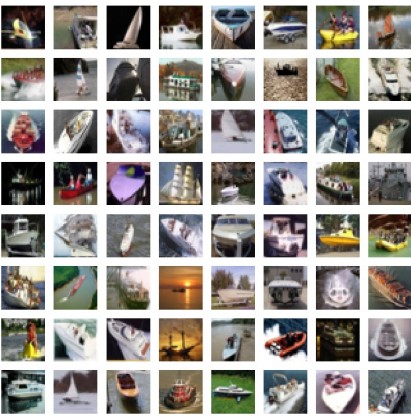

(q) Easiest examples of ship

(r) Hardest examples of ship

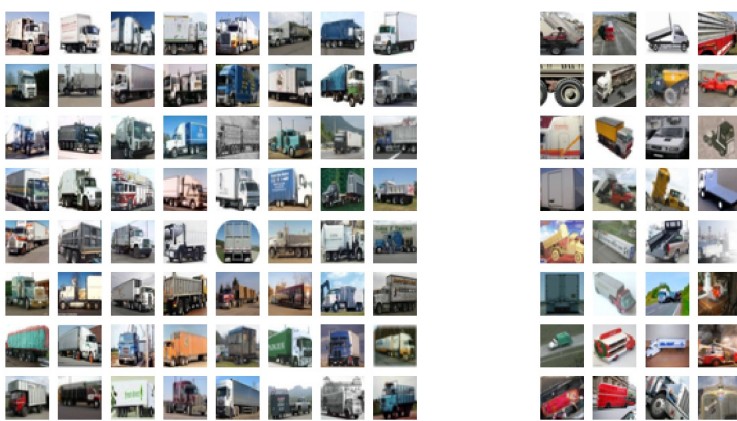

(s) Easiest examples of track                    (t) Hardest examples of track

Figure 6: Easy and hard examples of CIFAR-10. WRN 16-4 is used.

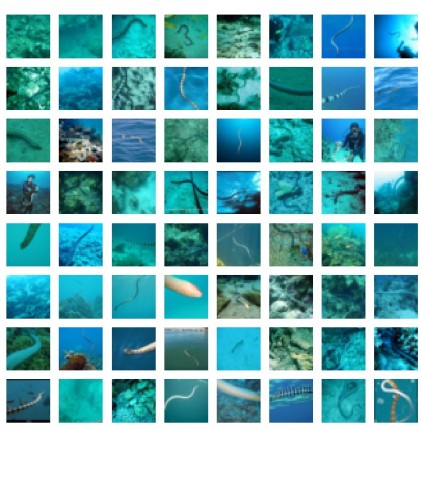
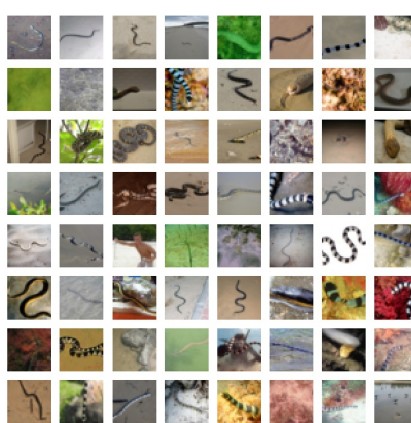

(a) Easiest examples of sea snake

(b) Hardest examples of sea snake

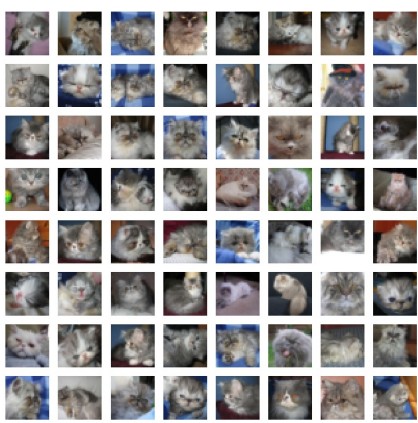
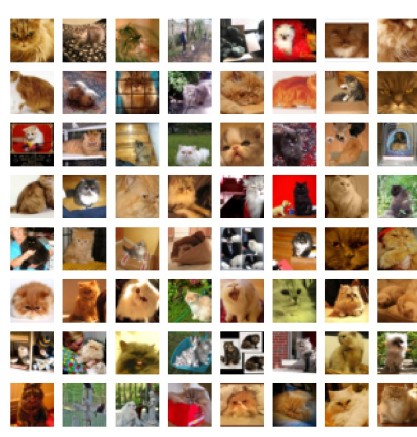

(c) Easiest examples of persian cat

(d) Hardest examples of persian cat

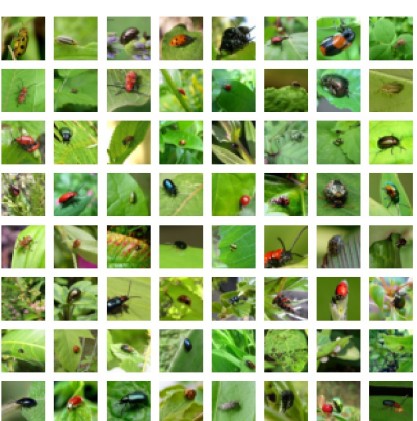
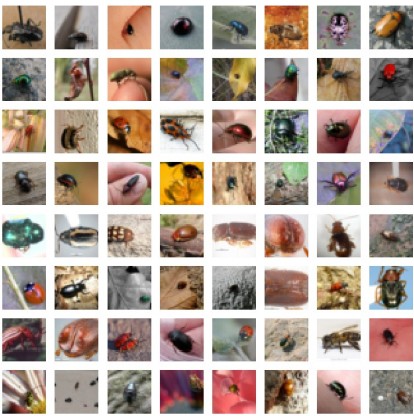

(e) Easiest examples of leaf beetle

(f) Hardest examples of leaf beetle

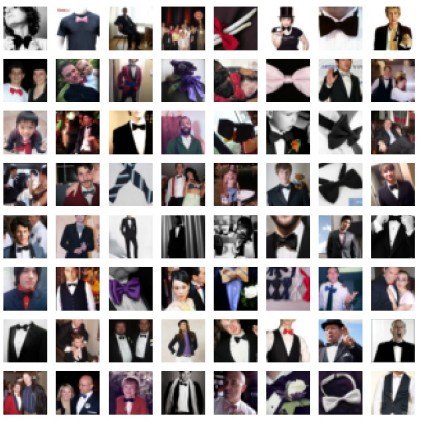

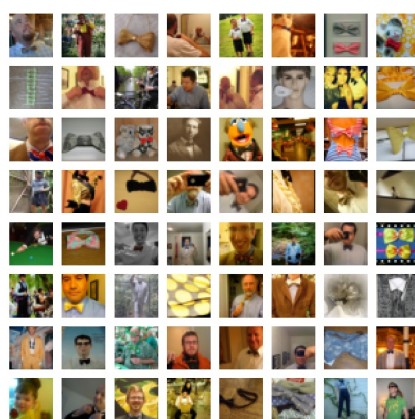

(g) Easiest examples of bow tie

(h) Hardest examples of bow tie

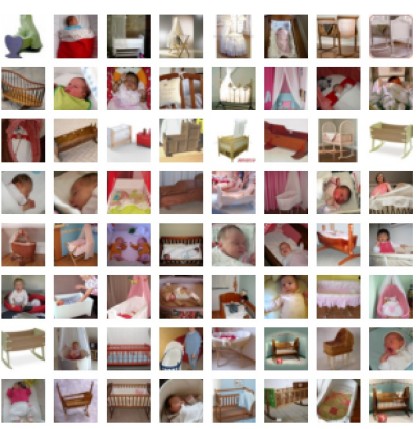

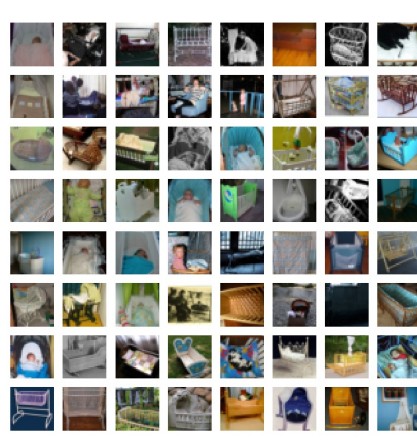

(i) Easiest examples of cradle

(j) Hardest examples of cradle

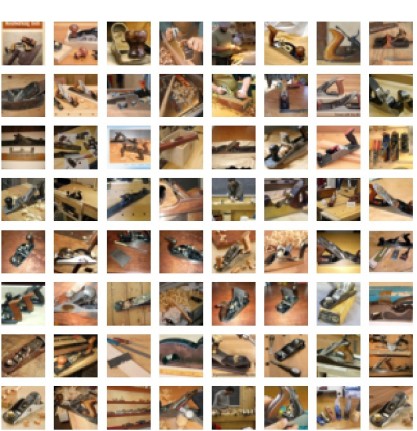

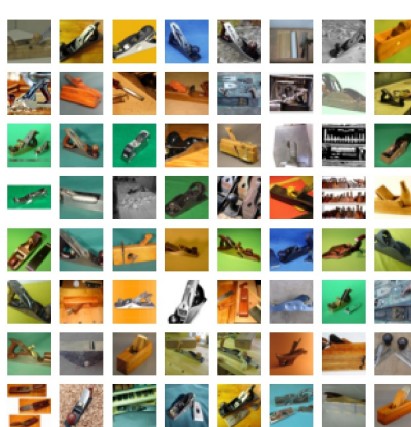

(k) Easiest examples of woodworking plane

(l) Hardest examples of woodworking plane

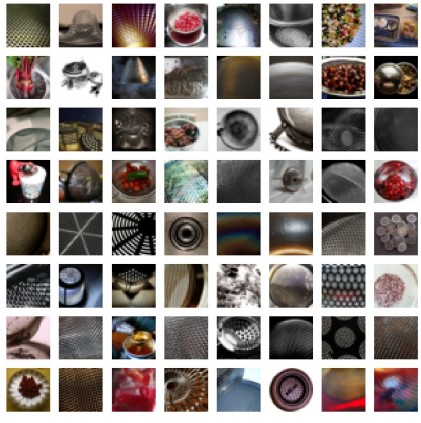

(m) Easiest examples of strainer

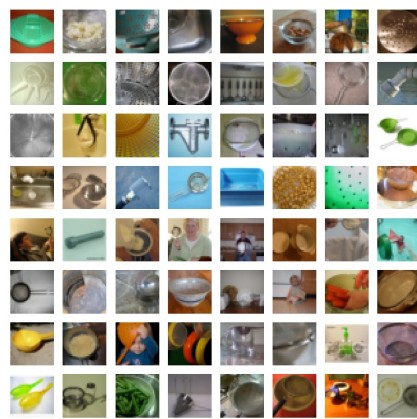

(n) Hardest examples of strainer

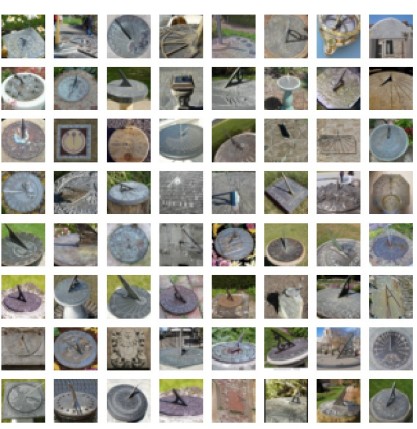

(o) Easiest examples of sundial

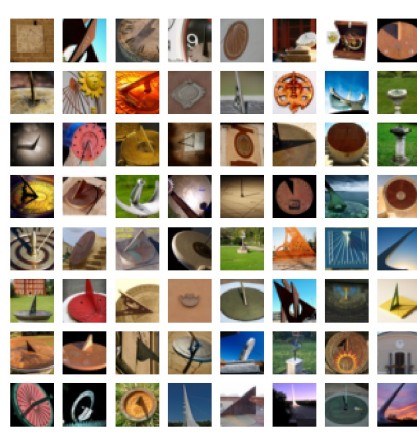

(p) Hardest examples of sundial

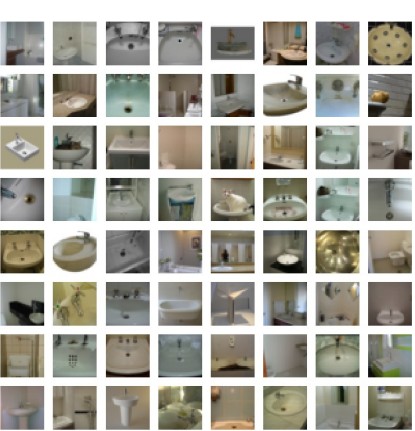

(q) Easiest examples of washbasin

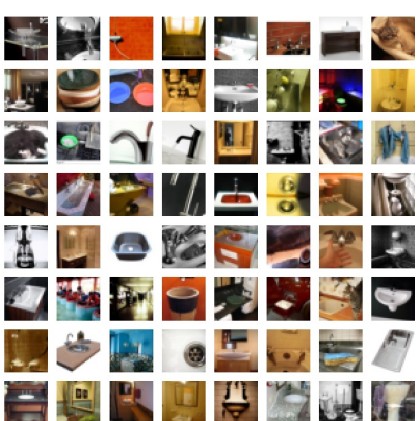

(r) Hardest examples of washbasin

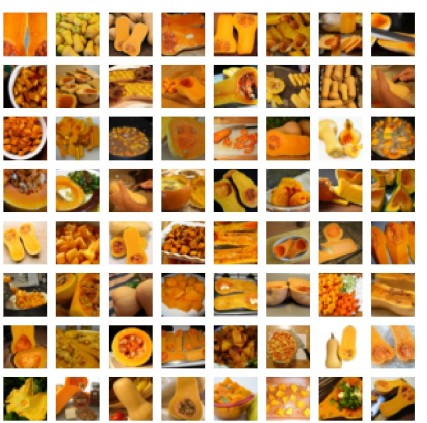 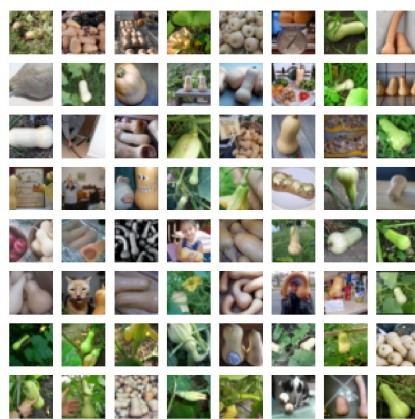

(s) Easiest examples of butternut squash      (t) Hardest examples of butternut squash

Figure 7: Easy and hard examples of mini ImageNet. ResNet 18 is used.

