# OpenReview forum: "Empirical Study of Easy and Hard Examples in CNN Training"
_ICLR.cc/2019/Conference_

### Official Review · AnonReviewer2 · 2018-10-28
**An experimental paper with many problems in the setting and analysis**

**Rating:** 3
**Confidence:** 4

**Review:**

This paper proposes a specific measure of difficulty for training examples called “easiness”. Easiness is based on training the model N times and counting the number of times an example is classified correctly. Based on this measure, they introduce “matching rate” as a measure of similarity of two architectures. Two architectures are suggested to be similar if the set of easy and hard examples is similar. The rest of the paper presents comparisons of architectures. Considering the problems below, I don’t see any reliable contribution in this paper.

- Why this specific definition of easiness? Can you compare to simply using “loss” as a measure for the difficulty of an example?
- e_t seems to be measuring the variance of training on a single example. If there is only one example that is always classified correctly, the denominator can be simplified to K. It doesn’t tell us how many training iterations it takes to fit that example.
- Why this specific formulation for “matching rate”? Why not a more common measure of similarity between sets such as intersection over union (IoU)? Can you suggest any references using a similar similarity score?
- Numbers in Table 1 do not seem particularly big to support the claim in section 4 that “...CNNs start learning from the *same* examples even if CNN architectures are different”. 0.20 is definitely bigger than random 0.1 for the matching rate but it still means only a 20% match.
- Random 0.1 is redundant in table 1.
- In section 4, define “contradicted patterns”.
- Are all images in Figure 1 for one model? How does it compare to visualizing examples according to their loss?
- The conclusion in section 5 says “... different CNNs start learning from similar patterns”. As mentioned above, “easiness” and consequently “matching rate” do not provide information about the progress of training and only final trained models. Regardless, this conclusion does not seem particularly unexpected or informative.
- Section 6 proposes to test a model on data with a different structure from data provided in training. This is a distribution mismatch and the model is not trained to handle.

---

### Official Review · AnonReviewer1 · 2018-10-31
**Interesting experiments but writing needs improvement**

**Rating:** 4
**Confidence:** 5

**Review:**

This paper extends the observation made in Arpit et al (2018) that deep networks prioritize learning simple patterns that are shared across many training examples. This paper further digs in this direction by defining a measure of "easiness" of a sample and making several empirical observations using this measure. This paper finds that the set of examples that can be identified as easy and hard for different architectures have a large intersection across architectures. It is also shown qualitatively that the set of easy examples have visually similar characteristics while hard examples are different from easy examples and also from one another. By artifically alterning the training images by changing color, structure and frequency components in individual examples, it is shown that the training process targets different characteristics that is dataset dependent. The effect of dropping samples from the set of easy and hard examples is studied and it is shown that dropping a small fraction of easy examples does not hurt generalization significantly (due to redundancy in patterns contained in such examples) while dropping even a smaller fraction of hard examples hurts generalization more significantly.

Overall i find the above empirical observations and some of the other arguments in this paper interesting but i think there is a lot of scope for improvement in the paper. In its current form, I find the language of this paper quite informal in a number of places, and some of the claims/deductions made in the paper are not well justified and they need to be changed accordingly. If these issues are fixed, I will improve my score. Specifically the issues are:

1. The notion of "easiness" introduced in Eq. (1) is clearly inspired by the experiment in Fig. 1 of Arpit et al (2018), and is also very similar. However, there is no mention of this in the paper.

2. Based on the experiments in table 1, it is mentioned in section 4 that "CNNs start learning from the *same* examples even if CNN examples are different." This statement is simply inaccurate. There is a decent fraction of easy and hard examples that are shared across different architectures in some cases. But this does not imply what the authors have claimed. The claim seems very careless.

3. In the sentence following "Why there are easy and hard examples?", the authors introduce the hypothesis of frequent patterns that are not *contradicted across classes*. I find this terminology unusual. What does it mean by patterns being contradiction? These sentences need to be re-worded.

4. Following this hypothesis, it is mentioned that "SGD force the model not to use contradicted patterns". Grammar aside,  the argument of SGD forcing the model sounds very informal. Even beyond that, I am not sure I see the basis for making such a claim. This whole paragraph (and the one that follows) sounds like a hypothesis that the authors have in mind about the observation made in table 1, but it is put forth as if they are facts.

5. In section 5, based on the experiment that shows that easy examples are visually similar, the authors write, "This result *implies* that different CNNs start learning from similar patterns.". This is again bad deduction. The qualitative results at best *suggest* that different CNNs start learning from similar patterns, but do not imply it.

6. The experiment in section 6 was interesting but the text was hard to follow and did not describe the results clearly.

7. In section 7, it is mentioned that "Randomly removing examples consistently produces the best performance." This sentence is misleading. Randomly dropping samples clearly hurts performance compared to baseline. The claim should be that randomly removing examples hurts the performance least compared with dropping easy and hard examples.

8. Finally, the authors mention at multiple instances that the observations made in this paper cannot be explained by the hypothesis set forth by Arpit et al (2018). I am not sure I understand the relevance of this claim. Explaining the observations made in this paper is never mentioned as a goal in Arpit et al (2018). These statements need to be fixed.

---

### Official Review · AnonReviewer3 · 2018-11-01
**Interesting premise, but analysis is shallow and offers little surprises**

**Rating:** 3
**Confidence:** 4

**Review:**

The paper formulates a definition of easy and hard examples and studies the properties and the training implications of such examples. The paper does not attempt to present insights that change training for the better (although suggests this could be future work), so the primary value it claims to add is our understanding of neural networks. I think the paper presents findings that most deep learning practitioners already find intuitive, which is why I think the paper falls short in its primary mission. An exposé like this could be valuable for an introductory text in deep learning, but I do not think the analysis meets the bar for a cutting edge insight and do not think it should be accepted.

Strengths:
- Quantifying easy and hard and using that as a starting point for further analysis is not a bad starting point at all.
- The experiments form a good starting point for interesting analysis.
- The paper is easy to follow and understand.

Weaknesses:
- The biggest weakness is that I just didn't find any of conclusions from this study to be that surprising or interesting. I think it's pretty obvious that neural networks start by learning the most immediately discriminative features ("frequent patterns"). The visual examples of easy and hard examples are not surprising at all (I think you get similar clusters if you just show bottom and top of model confidences, which I think many of us have). In section 7.1, the result that most misclassified examples are hard examples is presented as a surprising result. This is confusing, because this exactly what I would have expected given how you define easy/hard. It would be far more surprising if misclassified examples were all considered easy under your definition.
- The paper only scratches the surface. In Figure 2, the results for the two different datasets are quite different. This means only two datasets is probably not enough for us to understand what is going on here in general. Just the conclusion that datasets may be different in terms of easy/hard samples does not take the analysis far enough. It's also unclear what the reader should make of these conclusions.
- In Table 1, let's say the bottom 30% of samples are actually equally easy. This would mean that the "easy" examples are just a random 1/3 of those samples. Basically, I'm worried about the implications of having a hard cut-off at 10% and if there are situations where the bottom 10% actually changed quite a bit, but the broader picture of easy really didn't change that much. I guess I'm saying that I didn't quite gain confidence that definitions of easy/hard and matching rate are the correct way to go here and there might be a better metric that can look at the continuum of easy/hard from e = 0 to 1. You could have some kind of distance function where if an example moved from 5th percentile to 12th percentile, it would constitute a distance of 7. This is perhaps not the right thing either, but presenting an alternative metric and showing that the numbers (up to scale) and conclusions are unchanged would be nice.

Other comments:
- The new terminology of "contradicted pattern" and "non-contradicted pattern" is a bit confusing. Why aren't you just calling these "non-discriminative" and "discriminative"? If a mantis and a ladybird are both typically on a leaf, the leaf is not discriminative for this task. However, if a mantis and a boat are typically on differently colored backgrounds, the background is discriminative.

Minor comments:
- page 1, "easy and hard examples differ on various CNNs architectures" -> "CNN"
- page 2, "as a criteria" -> "criterion"
- page 2, "We then redefine easy and hard" -> don't you mean just "define"? Or do you mean that the words already have casual meanings, so this is a redefinition? I still think "define" is less confusing here.
- page 2, I think it's confusing that both easy and hard use the threshold \tau, suggesting it is the same. Maybe put a subscript to make it clear that the two \taus are different.
- page 6, "accuracy does not drop" -> could use a "does not *even* drop" for clarity
- page 7, "7.1 Do misclassified examples in validation dataset are hard examples": "Do"->"Are", remove "are"

---

### Meta-Review · Area_Chair1 · 2018-12-13

**Confidence:** 5
**Recommendation:** Reject

**Metareview:**

There is no author response for this paper. The paper formulates a definition of easy and hard examples for training a neural network (NN) in terms of their frequency of being classified correctly over several repeats. One repeat corresponds to training the NN from scratch. Top 10% and bottom 10%  of the samples with the highest and the lowest frequency define easy and hard instances for training. The authors also compare easy and hard examples across different architectures of NNs.
On the positive side, all the reviewers acknowledge the potential usefulness of quantifying easy and hard examples in training NNs, and R1 was ready to improve his/her initial rating if the authors revisited the paper.
On the other hand, all the reviewers and AC agreed that the paper requires (1) major improvement in presentation clarity -- see detailed comments of R1 on how to improve as well as comments/questions from R3 and R2; try to avoid  confusing terminology such as ‘contradicted patterns’.
R1 raised important concerns that the proposed notion of easiness is drawn from the experiment in Fig. 1 of Arpit et al (2017) which is not properly attributed. R3 and R2 agreed that in its current state the experimental results are not conclusive and often non informative. To strengthen the paper the reviewers suggested to include more experiments in terms of different datasets, to propose a better metric for defining easy and hard samples (see R3’s suggestions).
We hope the reviews are useful for improving the paper.